# Preparation of iron(IV) nitridoferrate $Ca_4FeN_4$ through azide-mediated oxidation under high-pressure conditions

Simon D. Kloß [1✉], Arthur Haffner[2], Pascal Manuel [3], Masato Goto[4], Yuichi Shimakawa [4] & J. Paul Attfield [1✉]

Transition metal nitrides are an important class of materials with applications as abrasives, semiconductors, superconductors, Li-ion conductors, and thermoelectrics. However, high oxidation states are difficult to attain as the oxidative potential of dinitrogen is limited by its high thermodynamic stability and chemical inertness. Here we present a versatile synthesis route using azide-mediated oxidation under pressure that is used to prepare the highly oxidised ternary nitride $Ca_4FeN_4$ containing $Fe^{4+}$ ions. This nitridometallate features trigonal-planar $[FeN_3]^{5-}$ anions with low-spin $Fe^{4+}$ and antiferromagnetic ordering below a Neel temperature of 25 K, which are characterised by neutron diffraction, $^{57}Fe$-Mössbauer and magnetisation measurements. Azide-mediated high-pressure synthesis opens a way to the discovery of highly oxidised nitrides.

[1] University of Edinburgh, Centre for Science at Extreme Conditions and School of Chemistry, Edinburgh EH9 3FD, UK. [2] Ludwig-Maximilians-University Munich, Department Chemistry, 81377 Munich, Germany. [3] ISIS Neutron Source, STFC Rutherford Appleton Laboratory, Oxfordshire, Didcot OX11 0QX, UK. [4] Institute for Chemical Research, Kyoto University, Uji, Kyoto 611-0011, Japan. ✉email: v1skloss@ed.ac.uk; j.p.attfield@ed.ac.uk

The search for unusual oxidation states fuels explorative inorganic chemistry for materials discovery[1]. In solid-state materials of 3d transition metals, the highest oxidation states are attained in oxides and fluorides, for example, $M_2Fe^{VI}O_4$ with $M$ = Li, Na and $Cs_2Cu^{IV}F_6$, while solid-state nitrides feature substantially lower states, especially with the later 3d metals as in $Li_3Fe^{III}N_3$ or $BaCu^IN$[2–6]. This disparity in observed oxidation states illustrates severe synthetic difficulties stemming from the physical properties of nitrogen. The low chemical potential of $N_2$ reflects the very stable $N\equiv N$ triple bond (944.8 kJ/mol) and nitrogen's relatively low electronegativity and positive first electron affinity (EA(N) = +0.07 eV compared to EA(O) = −1.46 eV), so nitrides generally have lower free energies of formation, less ionic bonding, and a tendency to lose $N_2$ at elevated temperatures[7–10]. As a result, the number of known 3d metal nitrides is small compared to that for oxides, despite much interest in their properties and applications[9]. Hence, high-throughput and data mining studies have been carried out to help discover unknown nitrides[9,11–13], and synthetic advances have been made by exploiting the increase in chemical potential of $N_2$ under pressure[14], as exemplified in the recent diamond anvil cell (DAC) synthesis of $MN_2$ ($M$ = $Ti^{IV}$, $Fe^{III}$ to $Cu^{I/II}$) pernitrides and $Fe^{II}N_4$ featuring polymeric nitrogen chains[15–19].

The large group of ternary transition metal nitrides $A_xM_yN_z$, in which A is an electropositive metal, also have a rich variety of electronic, physical, and magnetic properties[20–28], but their synthesis suffers from the limitation that the starting materials, e.g. binary transition metal nitrides or pure metals, usually feature lower oxidation states than the targeted compounds, for example, $Mn_4N$ vs. $Ca_6Mn^{III}N_5$[29]. As recently described in a review by Salamat et al.[30], the reactivity of $N_2$ at ambient to moderate pressures restricts access to high oxidation states, in particular for nitrides of the later 3d metals Mn to Cu[25,27,31,32].

In this work, we devise an adaptable synthesis route to highly oxidised nitridometallates by employing a standard large-volume press to generate high-pressure and high-temperature conditions with sodium azide $NaN_3$ as a powerful solid-state nitriding agent for use in a sealed reaction crucible. The method is illustrated by a straightforward preparation of the previously-unreported calcium nitridoferrate(IV) $Ca_4FeN_4$ starting from accessible binary reagents $Fe_2N$ and $Ca_3N_2$. While in molecular chemistry, high valent iron(IV, V, VI) nitrido complexes have already been investigated owed to their relation to biocatalytic and environmental catalytic processes[33–39], solid-state nitridoferrates prepared by direct combination reactions were reported with lower oxidation states +I to +III[5,40–42]. In the here presented reaction, sodium may act as a flux facilitating single-crystal growth and the large-volume-press enables the preparation of sufficient quantities of material (yield ca. 50 mg per experiment) for characterisation with $^{57}Fe$-Mössbauer spectroscopy, magnetometry, and neutron diffraction.

## Results

**Synthesis and chemical analysis.** $Ca_4FeN_4$ was prepared according to Eq. (1) at 6 GPa and ca. 1200 °C in a multianvil large-volume press and crystallises as a black microcrystalline powder containing plate-like crystals up to 30 μm in size (Supplementary Fig. 5):

$$8Ca_3N_2 + 3Fe_2N + 3NaN_3 \rightarrow 6Ca_4FeN_4 + 3Na + 2N_2. \quad (1)$$

As expected for nitridometallates, $Ca_4FeN_4$ is very sensitive to moisture and quickly hydrolyses forming the respective metal hydroxides and ammonia. To ensure a complete oxidation of $Fe_2N$, the reaction was carried out with a surplus of $NaN_3$.

Moreover, heating ramps of minimum 1 h were optimal for synthesis of pure samples, as faster heating led to large amounts of impurities, and dwell times of 5 h were required for single-crystal growth (see Supplementary Methods). The equation suggests Na formation, however, powder X-ray diffraction revealed unidentified byproduct(s) (Supplementary Fig. 3) but no Na metal. Energy dispersive X-ray (EDX) spectroscopy of the sample (15 datapoints, normalised on Ca, Supplementary Table 9) resulted in an experimental composition of $Ca_{4.0(4)}Fe_{1.1(1)}N_{4.1(6)}$, which fits the sum formula determined by single-crystal diffraction without detectable Na substitution. The incorporation of Na into the crystal structure by substitution of Ca would necessitate either ~50% of $Fe^V$ or ~25% $Fe^{VI}$ or additional incorporation of equal amounts of O to retain $Fe^{IV}$. These can be ruled out as $^{57}Fe$-Mössbauer data show only one isomer shift at large negative velocity and neutron powder diffraction refinement of the N occupancy ruled out additional incorporation of O (see later sections). EDX revealed Na with a fraction of up to 5 at-%, close to the theoretical Na content from the reaction equation (5.5 at-%). This might either be finely dispersed sodium metal, undetected in the PXRD data, or a constituent of a Na–Ca/Fe–N byproduct.

**Structure discussion.** The crystal structure of $Ca_4FeN_4$ (space group *Ibca*, a = 6.903(2), b = 6.919(3), and c = 22.552(8) Å) was solved and refined from single-crystal X-ray diffraction data. Details are in the Supplementary Discussion. The structure contains three Ca and one Fe position and can be described by a relatively dense packing of face-, edge-, and corner-sharing distorted $CaN_6$ octahedra (Fig. 1a,b), in which trigonal-planar $[Fe^{IV}N_3]^{5-}$ units (Fig. 1c) are embedded via edge-sharing. Fe forms a double-layered quasi-2D tetragonal sublattice (Supplementary Fig. 1) with short Fe–Fe distances (4.89 and 4.93 Å) within the layers and long ones (8.49 Å) between. The crystal field exerted on the $Fe^{4+}$ ions (qualitatively displayed in Fig. 1d) is expected to result in low-spin configuration with $S = 1$ owing to strong $d_\pi$–$p_\pi$ Fe–N bonding in accordance with existing studies on monoatomic double-faced π-donors in general and on nitrido-ligands particularly[43–45]. A detailed discussion of the structure, d-orbital splitting and spin-state of Fe are enclosed in the Supplementary Discussion, while experimental verification follows in the later sections through magnetisation, $^{57}Fe$-Mössbauer, and neutron diffraction measurements.

As no electronic instability exists, the slight distortion towards Y-conformation in the $[FeN_3]^{5-}$ complex anion (Fig. 1c, $d_{Fe–N}$ = 1.731(6), 1.731(6), 1.758(7) Å, Fe–N–Fe angles 121.1(2)°, 121.1(2)°, 117.8(4)°) probably stems from the surrounding crystal structure rather than a Jahn-Teller distortion[44]. $Ca_4FeN_4$ exhibits shorter Fe–N bond lengths than the known Ca nitridoferrate(III) $Ca_6Fe^{III}N_5$ with $d_{Fe–N}$ = 1.769(15) Å but similar bond lengths are observed in nitridoferrates(III) of the higher homologues, $Sr_3FeN_3$ and $Ba_3FeN_3$ with $d_{Fe–N}$ = 1.73(1) Å, which feature a larger inductive effect, shortening the Fe—N bonds[42,46,47]. The range of observed Ca–N distances (2.43 < $d_{Ca–N}$ < 2.82 Å, Supplementary Fig. 2 and Supplementary Table 4) is in good agreement with values found in other Ca nitridoferrates like $Ca_2FeN_2$ and $Ca_6FeN_5$ (2.33 < $d_{Ca–N}$ < 2.74 Å) and Ca nitrido-metallates like $CaTiN_2$, $Ca_4TiN_4$, and $Ca_3CrN_3$ (2.37 < $d_{Ca–N}$ < 2.85 Å)[20,23,41,47,48].

Bond valence sum (BVS) calculations (Supplementary Table 10) led to calculated valences of $V_{Fe1}$ = 4.15, $V_{Ca1}$ = 2.08, $V_{Ca2}$ = 1.85, $V_{Ca3}$ = 1.65 corroborating a high valence for the iron atom but also showing that Ca2 and Ca3 are underbonded, which might be related to metal-metal second nearest neighbour

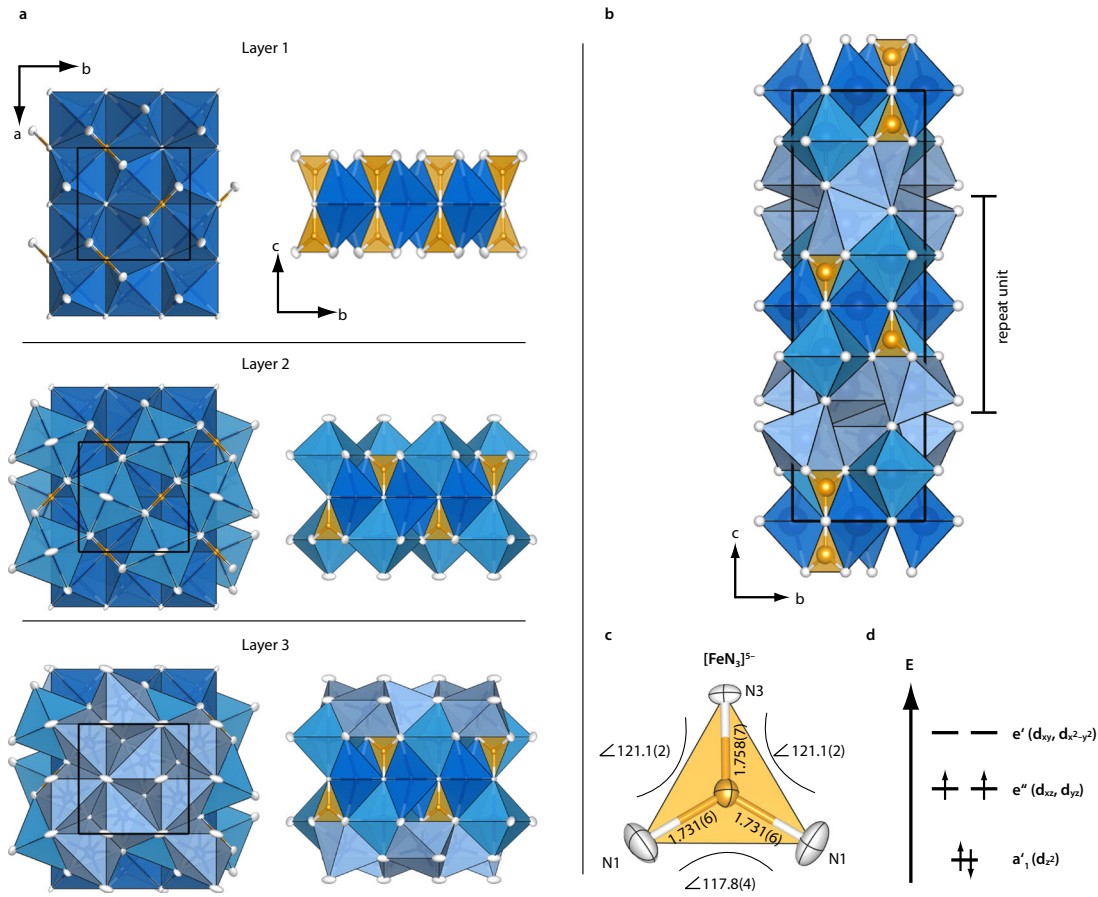

**Fig. 1 Structure of Ca₄FeN₄. a** Formal construction of the repeat unit, which can be subdivided into three layers. Layer 1 is formed by Ca1N₆ octahedra and [FeN₃]⁵⁻ polygons, layer 2 is formed by Ca2N₆ octahedra, and layer 3 by Ca3N₆ octahedra. The repeat unit is created by stacking of the layers above and below layer 1, with an inversion centre in the middle of layer 1. **b** Structure of $Ca_4FeN_4$ with inequivalent Ca positions coloured differently, highlighting the repeat unit. **c** [FeN₃]⁵⁻ complex anion showing bond lengths (in Å) and N–Fe–N bond angles (in deg.). Ellipsoids at 95 % probability. **d** Qualitative d-orbital splitting vs. energy for the [FeN₃]⁵⁻ complex anion with double-faced π-donor N-ligands and $d^4$ electron configuration with $S = 1$ (adapted from literature)[43].

interactions between closely situated Ca2 and Ca3 atoms ($d_{Ca2–Ca2} = 2.997(3)$ Å, $d_{Ca2–Ca3} = 3.129(2)$, $3.272(2)$ Å)[49,50]. Further discussion of the bonding situation can be found in the Supplementary Discussion.

**Magnetometry**. Susceptibility measurements on $Ca_4FeN_4$ were carried out in a field of 30 kOe with zero-field-cooling for the range from 2 to 300 K and also in field-cooled mode from 2 to 50 K (Fig. 2). The susceptibility curve shows paramagnetic behaviour down to approximately 50 K and a sudden decrease at 25 K consistent with an antiferromagnetic transition. The upturn observed below 7 K is likely a Curie tail from a paramagnetic impurity, with effective moment <0.3 μ_B (see Supplementary Discussion). A Curie–Weiss fit to the paramagnetic regime of the susceptibility from 80 to 300 K resulted in an effective paramagnetic moment of $\mu_{eff} = 3.08(1)$ μ_B and a Weiss-constant of $\Theta = −123(1)$ K. The negative value of the Weiss-constant is consistent with the observed antiferromagnetic ordering. However, the observed transition temperature is much lower than $|\Theta|$, which is possibly due to low magnetic dimensionality originating from the quasi-2D tetragonal Fe sublattice (Supplementary Fig. S1). The observed paramagnetic moment is slightly greater than the ideal spin-only value of 2.83 μ_B for a spin $S = 1$ system, in keeping with the 2.95 μ_B value observed for a hexahydrazide cage iron(IV) complex, and hence corroborates the low-spin state in the trigonal-planar [Fe^IVN₃]⁵⁻ units[33].

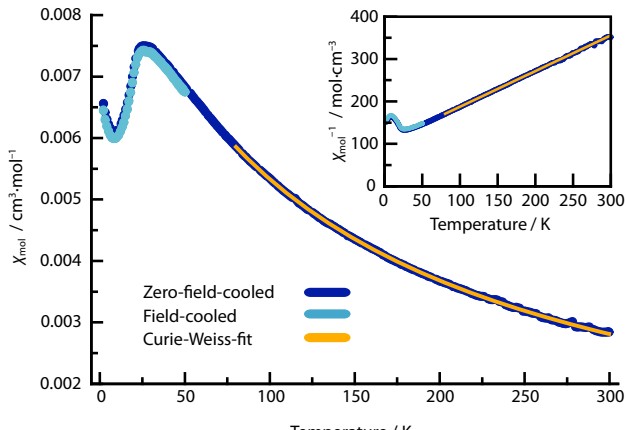

**Fig. 2 Temperature dependent molar susceptibility χ_mol of Ca₄FeN₄.** Zero-field cooled and field cooled data are shown. The Curie–Weiss fit was performed for the temperature range of 80 to 300 K. The inset shows that inverse molar susceptibility $\chi_{mol}^{-1}$ varies linearly above 75 K, highlighting that a single dominant Curie–Weiss paramagnetic phase is present.

Isothermal magnetisation curves (Supplementary Fig. 6) show only a small hysteresis due to impurities, equivalent to ~0.01% Fe metal, and confirm that the ground state of $Ca_4FeN_4$ is antiferromagnetic.

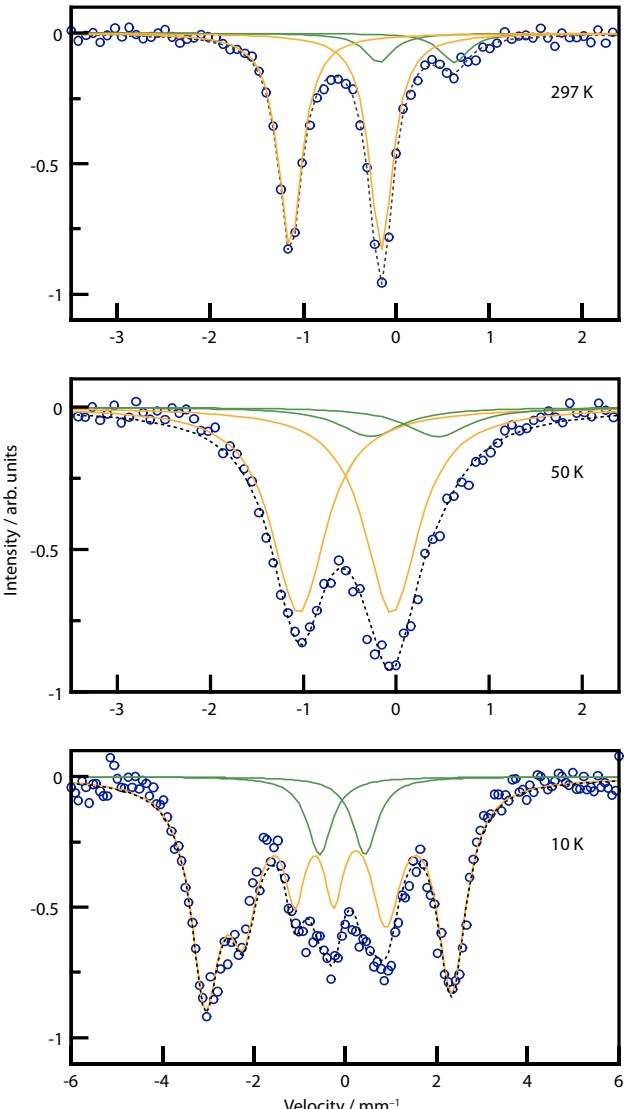

**Fig. 3 $^{57}$Fe Mössbauer spectra of Ca$_4$FeN$_4$.** Data (blue circles) were collected at 297, 50 and 10 K, orange curve belongs to Fe$^{IV}$ of Ca$_4$FeN$_4$, green curve belongs to an unidentified byproduct, grey dashed line is the sum of the individual curves. Green byproduct doublet appears at isomer shifts $\delta_{297K} = 0.22$, $\delta_{50K} = 0.08$, $\delta_{10K} = -0.6$ mm/s (intensity ratio Fe$^{4+}$/Fe$_{impurity}$ = 85.8/14.2 was fixed in the low temperature fits using the 297 K value).

**Mössbauer spectroscopy.** To further corroborate the Fe$^{IV}$ oxidation state, $^{57}$Fe-Mössbauer spectra of Ca$_4$FeN$_4$ (Fig. 3) were collected at 297, 50, and 10 K. The RT and 50 K spectra, which are above the antiferromagnetic transition temperature, show doublets at large negative velocity of $\delta_{297K} = -0.65$ mm/s and $\delta_{50K} = -0.55$ mm/s. Similar isomer shifts have been observed in Fe$^{V}$ containing oxides like La$_2$LiFeO$_6$ ($\delta = -0.41$ mm/s) and Fe-doped K$_3$MnO$_4$ ($\delta = -0.55$ mm/s)[51,52]. The isomer shift is, however, strongly related to covalency of the Fe–ligand bond as well as the coordination environment, both influencing the s-electron density at the nucleus[53]. Ab initio calculations on Ba$_3$[Fe$^{III}$N$_3$] with $\delta_{263K} = -0.55$ mm/s of Fe$^{III}$ showed that trigonal-planar coordination leads to strong covalent σ-type Fe–N bonding filling the 4s-orbital while d$_\pi$–p$_\pi$ backbonding depletes the electron density in the d-orbitals resulting in a reduced shielding effect[45]. The observed isomer shift in Ca$_4$FeN$_4$ of $\delta_{297K} = -0.65$ mm/s can thus be rationalised from the higher oxidation state Fe$^{IV}$ as well as

**a**

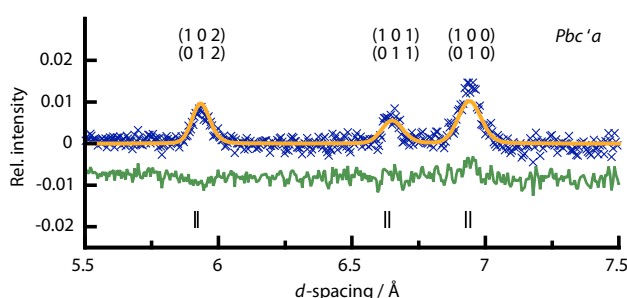

**b**

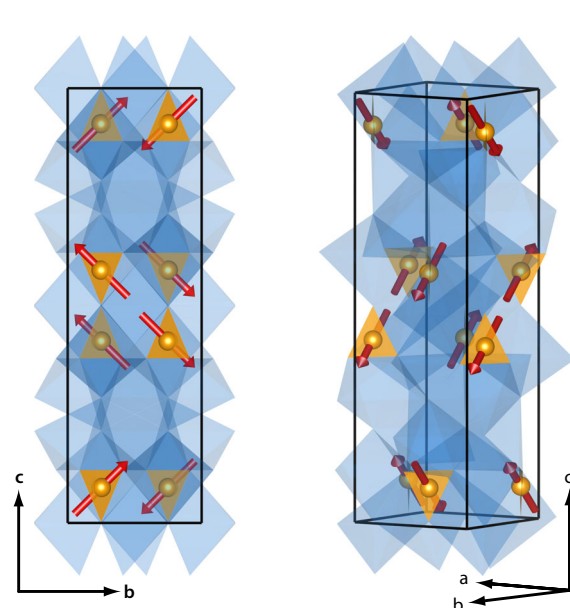

**Fig. 4 Magnetic structure of Ca$_4$FeN$_4$. a** Magnetic structure refinement carried out on the difference of data collected at 1.5 and 50 K of detector bank 1. Blue crosses are datapoints, orange is the fit of the magnetic structure, green is the difference curve. Positions of reflections are marked by vertical drop lines, their *hkl* indices displayed above the reflection. **b** Magnetic structure shown in two different directions. [FeN$_3$]$^{5-}$ polygons in orange, CaN$_6$ polyhedra in blue.

a similar bonding situation as in Ba$_3$[Fe$^{III}$N$_3$]. The oxidation state Fe$^{IV}$ in nitrides has only previously been reported in mixed Fe$^{III}$/Fe$^{IV}$ materials Li$_{3-x}$[FeN$_2$] observed during topotactic electrochemical redox deintercalation of Li$_3$[Fe$^{III}$N$_2$] for which in operando Mössbauer spectroscopy revealed isomer shifts of up to $\delta = -0.33$ mm/s[54].

The quadrupole splitting observed in the 297 K and 50 K spectra of Ca$_4$FeN$_4$ ($\Delta_{297K} = 0.98$ mm/s, $\Delta_{50K} = 1.00$ mm/s) is expected for the non-cubic coordination environment. At 10 K, the Mössbauer spectrum shows sextet peaks in line with the antiferromagnetic ordering at 25 K observed in the susceptibility data. The fit of the sextet resulted in $\delta_{10K} = -0.52$ K, $\Delta_{10K} = 0.31$ mm/s and a hyperfine splitting of 16.7 T. The increase in isomer shift with decreasing temperature is probably owed to the second-order Doppler effect[53].

**Neutron diffraction.** Neutron diffraction patterns were collected at 50 and 1.5 K, above and below the 25 K antiferromagnetic transition temperature of Ca$_4$FeN$_4$. A low temperature nuclear structural model was established by Rietveld refinement of the 50 K data and then transferred to the 1.5 K data (Supplementary Fig. 4, Supplementary Table 7). Atom positions were freely refined, and displacement parameters of equal atom types were constrained.

Tentative refinement of possible N/O mixed anion positions indicated complete N-occupation. Results of the refinements are summarised in the Supplementary Discussion.

Three magnetic reflections are observed in the diffraction pattern obtained at 1.5 K (Fig. 4a). The difference 1.5–50 K diffractogram was used for magnetic structure refinement as one magnetic reflection overlapped with a small reflection from a byproduct. The magnetic reflections were indexed on a unit cell of the same dimensions as the nuclear one but with space group $Pbca$ (space group no. 61), which is a maximal subgroup of the nuclear model's space group $Ibca$ (no. 73). The best fit was achieved with refinement in magnetic group $Pbc'a$ (a non-standard setting of $Pb'ca$) of the crystallographic Bravais-class (see Supplementary Discussion). The spins of the resulting antiferromagnetic structure (Fig. 4b) are oriented parallel to the crystallographic $bc$-plane and the observed magnetic moment $\mu = 1.92(10)$ $\mu_B$ (components $\mu_y = 1.36(10)$ $\mu_B$, and $\mu_z = 1.36(3)$ $\mu_B$) is in good agreement with the ideal saturated moment of $2S = 2$ $\mu_B$.

We report the synthesis of $Ca_4FeN_4$ with Fe in high oxidation state +IV via high pressure and high-temperature azide-mediated oxidation and its characterisation through $^{57}Fe$-Mössbauer, magnetisation, and neutron diffraction measurements. High pressures may be a necessity for the synthesis of $Ca_4FeN_4$ as bond valence sum calculations indicated metal-metal repulsion between adjacent Ca atoms, which might hinder the formation of this structure at lower pressures. At ambient pressure, the structure might be stabilised through the electron inductive effect of Ca. The crystal field splitting in the $[FeN_3]^{5-}$ anions do not result in an electronic instability in the low-spin $Fe^{4+}$ d-orbitals and thus probably are stabilised with respect to the disproportionation into $Fe^{3+}$ and $Fe^{5+}$, which is observed in $CaFeO_3$ and other high valent iron perovskites[55]. Susceptibility and neutron diffraction measurements do not suggest the presence of ligand holes ($d^5\underline{L}$) as often observed for $Fe^{IV}$ compounds[56], which might be destabilised through lower lying p-orbital levels of N when compared to O.

The formation of $Ca_4FeN_4$ seemingly resembles a Na-flux reaction as crystallisation is facilitated through longer reaction times but the oxidation mechanism to $Fe^{4+}$ is in question[32]. Achieving this high oxidation state certainly requires a strong nitriding agent, however, the behaviour of $NaN_3$ under high pressure and temperature conditions has not been elucidated. $NaN_3$ could decompose into Na and $N_2$ or the pressure could prevent the decomposition and lead to the stabilisation of the highly reactive azide itself or its partial decomposition product the diazenide, as reported for decomposition of $M(N_3)_2$ ($M = Ca$, Sr, Ba) under pressure[57].

The investigation of the here described oxidation process will probably require in situ experiments. However, the already applicable advantages of this route for the synthesis of nitrides are a highly nitriding environment in a large volume press, Na probably acting as a flux facilitating crystal growth, and large sample quantities available for property characterisation. Moreover, it does not rely on specific properties like the ability to deintercalate Li, which gave the only previous report of a $Fe^{4+}$ nitride in the $Li_{3-x}[FeN_2]$ system[54]. The $NaN_3$-route overcomes the challenging thermodynamics of nitride chemistry and enables the direct synthesis of a highly oxidised transition metal nitride, which was not obtained from standard ambient and medium pressure methods nor with DACs at extreme $N_2$ pressures (>30 GPa) and temperatures (>2000 K)[19,27,32]. This route is thus expected to significantly advance nitride research by accessing new materials in high transition metal oxidation states and consequent new chemical and physical properties.

## Data availability

Single-crystal data are available through the joint CCDC/FIZ Karlsruhe inorganic crystal structure database by quoting number CSD 2015297.

The data that support the findings of this study are available at: https://doi.org/10.7488/ds/2935.

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

## Acknowledgements
The authors gratefully acknowledge Dr. Tobias Rackl (LMU Munich) for PPMS measurements. Funding was provided by a DFG research fellowship for Dr. S. D. Kloß and EPSRC and access to the ISIS facility is through STFC.

## Author contributions
Concept and idea for method by S.D.K. Synthesis, X-ray and neutron structure analysis, physical measurements data analysis performed by S.D.K. with J.P.A. as support. Neutron diffraction performed by P.M. and supporting role in data analysis. $^{57}Fe$-Mössbauer spectroscopy and data analysis performed by M.G. and Y.S. X-ray diffraction and SEM/EDX measurements performed by A.H. Manuscript written by S.D.K. with contributions from all authors and support from J.P.A.

## Competing interests
The authors declare no competing interests.
