## [Peer Review File · Nature Communications]

REVIEWER COMMENTS

Reviewer #1 (Remarks to the Author):

This is an interesting paper reporting the synthesis of a new type of crystalline ternary nitride in the Ca-Fe-N system using high pressure-high temperature (HPHT) techniques by chemical reaction from highly nitrated precursors. The structure determined by single crystal X-ray diffraction is quite beautiful, and the authors have determined the magnetic ordering using neutron diffraction and magnetic susceptibility measurements. The paper is definitely suitable for publication in *Nature Communications*.

The authors note that the formal oxidation state of Fe is unusually high (nominally 4+ assigning formal oxidation numbers to the Ca²⁺ and N³⁻ ions). Although high metal oxidation states (OS) are achieved readily among transition metal oxides and fluorides, using HPHT techniques to form some of the highest OS among Fe compounds, they are not typically encountered among nitrides that tend to show greater covalency in their bonding. In addition, as noted by the authors, the energetically favourable formation of N₂ gas poses a problem for syntheses at lower pressure. This area of research was recently reviewed by Salamat et al (*Coord Chem Rev* 257 (2013) 2063) and the authors might wish to reference this work in their introduction.

Because this is such an unusual structure type, application of "traditional" ionic models typically used to achieve rationalisation of crystal structures and why they exist does not provide useful information here. The analysis of the magnetic properties in terms of a crystal field model for the Fe⁴⁺ centres achieves its goal, but the statement that the crystal field splitting provides the energetic driving force for stabilisation of the crystal structure is less convincing.

I suggest that the authors first examine the extensive review of nitride crystal structures published by Nate Brese and Mike O'Keeffe as "Crystal Chemistry of Inorganic Nitrides", in the *Structure and Bonding* book series by Springer Berlin, 1992, pp. 307-378 (Brese N.E., O'Keeffe M. (1992) *Crystal chemistry of inorganic nitrides*. In: *Complexes, Clusters and Crystal Chemistry*. Structure and Bonding, vol 79. Springer, Berlin, Heidelberg. <https://doi.org/10.1007/BFb0036504>). This can be a bit difficult to source and one of the authors may have to purchase the chapter, but it is well worthwhile. The review examines many different nitride crystal structures critically and systematically (even highlighting previous mistakes in crystallographic structure determinations), using the principles of "bond valence" approaches to determining and studying the oxidation number (distinct from the formal OS) combined with concepts of metal-metal 2nd-nearest neighbour interactions, to give rise to the principle of "eutaxy". In addition, the authors highlight the principle of considering metal-only sublattices with inserted (much smaller) anions, to interpret many complex structure types. Reading and considering the possibilities of applying the concepts reviewed in this work are recommended here.

Next, using the bond valence systematics first described by ID Brown and published in an extensive table by Brese and O'Keeffe (*Acta Cryst B* 47 (1991) 192-197) could be very helpful to provide a more detailed picture about what is happening around the different cation sites. I have done a quick calculation using the Brese/O'Keeffe parameters for Fe-N and Ca-N bonds along with the determined bond lengths, that is very revealing. The total valence achieved at the trigonal Fe sites surrounded by N1 and N3 nitrogen atoms is 4.148, confirming the ionic model assignment to a 4+ state. The Ca1 site achieves a valency of 2.08, close to ideal. However, the Ca2 and Ca3 sites are seriously underbonded (1.83 and 1.63 respectively). This can not simply be ascribed to the greater "bonding power" of the Fe atoms pulling the N atoms closer to them, because the N2 atoms are not bonded to Fe. Instead, this most likely reflects the influence of metal-metal repulsions between the large Ca atoms sharing these N2 centres. In fact, achieving this structure under high pressure conditions might have more to do with overcoming Ca...Ca repulsions than satisfying crystal field energetic considerations. I would urge the authors to consider these possibilities that fall outside the normal, more "comfortable" ionic view of the crystal structures and how they "work". That could make an even more valuable contribution to the overall field of solid state chemistry and structure-properties design.

Finally, the authors note correctly that the main effect of pressure might not be to increase the thermodynamic activity of nitrogen to achieve the highly nitrated compound. In fact, although the melting vs thermal decomposition relations of NaN_3 are not yet known under HPHT conditions, it is established that Ca_3N_2 acts as a stable melt component in $\text{Li}_3\text{N}-\text{Ca}_3\text{N}_2-\text{P}_3\text{N}_5$ compositions that readily quenched to glass after heating to $\sim 1400^\circ\text{C}$ at 1-2 GPa pressures (Grande et al Nature 369 (1994) 43). It is entirely possible, and even likely, that the $\text{Ca}_3\text{N}_2-\text{NaN}_3$ mixture did not decompose completely, leading to a high N_2 gaseous/fluid within the high pressure reaction chamber, that provided a constant volume environment with a suitable flux for single crystal growth. In further work, the authors should examine the thermodynamic stability of their particular crystal structure and stoichiometry relative to other binary and ternary phases, because it is highly likely that what they have produced is a thermodynamically stable structure under this combination of HPHT conditions.

Reviewer #2 (Remarks to the Author): see attached

Reviewer #3 (Remarks to the Author):

The manuscript presents a new synthetic method for preparation of transition metal nitrides which enables formation of high oxidation states on the metal. A new compound is made in this manner and characterized using magnetic susceptibility, Mossbauer spectroscopy, and X-ray and neutron diffraction to determine its structure and the nature of the iron. This work is of interest to a broad audience as it covers a new synthesis technique as well as the properties of a new complex material. I would recommend publication if the following issues are addressed.

1. The paper repeatedly highlights the fact that the use of the large volume press allows for formation of fairly large quantities of product. But the amounts of reactants used and product obtained are not mentioned anywhere in the manuscript (not even in supporting information). In order to get an idea of how advantageous this method is, the reader needs to know more about the scale of your reactions!
2. The authors assume that the NaN_3 thermally decomposes to form sodium metal which acts as a metal flux. However, NaN_3 is not used in very large excess, and the reaction temperature is well above the boiling point of sodium at atmospheric pressure; is anything known about the state of sodium at 1200 C and 6 GPa? No sodium metal is seen in PXRD of the product but the authors state that the EDS shows "finely dispersed" sodium in the sample. Please add more information to this section--is this sodium in the elemental form, or is it part of a compound ("Na-Fe-N byproduct")?

The present manuscript, **NCOMMS-20-32258-T**, submitted by Kloß, Attfield and co-workers, reports, as properly titled, the “*Preparation of iron(IV) nitridoferrate Ca_4FeN_4 through azide-mediated oxidation under high-pressure conditions.*” The newly synthesized, solid-state material was structurally characterized by single-crystal x-ray diffraction analysis, and the compound’s electronic structure was studied by neutron diffraction and SQUID magnetization as well as ^{57}Fe -Mössbauer spectroscopy. Taken all together, the study reports a ternary nitride with a tetravalent iron ion.

This reviewer appreciated reading and learning about this study, and, ultimately, would recommend publication of this work in *Nature Communications*. However, the present version can certainly not be recommended for publication in this journal. As is, the study is clearly written for a very specific community in the field of inorganic chemistry, namely solid-state inorganic chemistry; and hence, is of very little interest to the particularly broad readership of this journal. This is a pity as the general topic of high-valent iron species is of great interest to bioinorganic and inorganic coordination chemists, spectroscopists and computational chemists.

That said, a revised version must place this work in context to those neighboring fields of inorganic chemistry!

Once the authors bothered to study those neighboring fields, they may find out that the iron 4+ oxidation state is not an unusual iron oxidation state *per se*.

In molecular coordination chemistry, K. Meyer and J. Smith reported the first structurally characterized Fe(IV) nitrides in 2008 (published in *Angewandte* and *JACS*) and the same authors published an Fe(V) nitrido complex in *Science* in 2011. Fleeting intermediates of Fe(IV), Fe(V) and even Fe(VI) nitridos were spectroscopically characterized by J. Peters (*JACS* 2004) and K. Wieghardt (e.g. *JACS* 1999, *Science* 2006). Noteworthy, I. O. Fritsky published an “indefinitely stable Fe(IV) cage complexes” in *Nature Communication*, **2016**. These studies must be mentioned and cited, and while this reviewer acknowledges the distinctiveness of the Fe(IV) center in Ca_4FeN_4 , the authors ought to tone down their “unusual” and “high” oxidation state. Further revisions should include the following:

The abstract is badly written. Why are high oxidation states difficult to stabilize due to the high thermodynamic stability and inertness of dinitrogen. I understand what the authors meant to say, but, as written, it makes no sense. Also, what is a “reactive nitrogen species.” In addition, the sentence is grammatically wrong, and the last sentence of the abstract reads bad, at best.

On page 4, line 61, the authors state that sodium azide, NaN_3 , is a “powerful solid-state nitriding agent.” However, the authors do not know where the nitride nitrogen atoms in their title compound stem from! This is a severe deficiency and ought to be addressed; possibly, by the use of ^{15}N -labeled NaN_3 . In fact, how can the authors rule out that the nitrides in their $[\text{Fe(IV)N}_3]^{5-}$ moiety do not originate from the precursor materials, Ca_3N_2 and Fe_2N ?

On page 4, line 66, the authors continue and state that “Sodium, which is released upon reaction, acts as a metallic flux...” This implies that NaN_3 is thermally decomposed to Na and N_3 radicals. The latter recombine and decompose to N_2 . This is literature-known and should be mentioned here. Whether or not the high-pressure conditions change this decomposition pathway is purely speculative.

Throughout the text, and the SI, when talking about Moessbauer spectroscopy, the authors must specify the isotope. Clearly, it is “ ^{57}Fe -Moessbauer” spectroscopy. Further, in Moessbauer spectroscopy, it is not the “chemical shift” (e.g. page 5, line 90) but the “isomer shift.”

Page 6, line 106/107: the “low-spin configuration” should be further specified as $S = 1$.

Page 6, line 115: the individual and average Fe–N bond distances should be mentioned/listed in the main text; e. g., page 6, line 115.

Page 8, line 143: The authors report the Weiss constant and the effective moment, μ_{eff} , to be $3.08(1) \mu_{\text{B}}$, and state that “the paramagnetic moment is close to the theoretical spin-only value of a system with $S = 1$ of $2.83 \mu_{\text{B}}$ and hence corroborate the low-spin state in the...”

This reviewer would appreciate a plot of μ_{eff} vs. T , and wonders

- a) how the experimental error “(1)” is determined,
- b) and how can the authors make this statement, if the ^{57}Fe -Moessbauer spectrum reveals an obvious and significant Fe-impurity (which, very likely, will affect the effective magnetic moment).

That said, the Fe-impurity (Fig. 3) must be quantified !

Response Letter

Manuscript no.: NCOMMS-20-32258-T

Publication title: Preparation of iron(IV) nitridoferrate Ca_4FeN_4 through azide-mediated oxidation under high-pressure conditions

Authors: Simon D. Kloß*, Arthur Haffner, Pascal Manuel, Masato Goto, Yuichi Shimakawa, J. Paul Attfield*

Reviewer #1:

This is an interesting paper reporting the synthesis of a new type of crystalline ternary nitride in the Ca-Fe-N system using high pressure-high temperature (HPHT) techniques by chemical reaction from highly nitrated precursors. The structure determined by single crystal X-ray diffraction is quite beautiful, and the authors have determined the magnetic ordering using neutron diffraction and magnetic susceptibility measurements. The paper is definitely suitable for publication in Nature Communications.

Our response: Thank you for evaluating our manuscript. We revised our manuscript according to your comments as follows, which improved the quality of our contribution.

The authors note that the formal oxidation state of Fe is unusually high (nominally 4+ assigning formal oxidation numbers to the Ca^{2+} and N^{3-} ions). Although high metal oxidation states (OS) are achieved readily among transition metal oxides and fluorides, using HPHT techniques to form some of the highest OS among Fe compounds, they are not typically encountered among nitrides that tend to show greater covalency in their bonding. In addition, as noted by the authors, the energetically favourable formation of N_2 gas poses a problem for syntheses at lower pressure. This area of research was recently reviewed by Salamat et al (Coord Chem Rev 257 (2013) 2063) and the authors might wish to reference this work in their introduction.

Our response: Done.

Because this is such an unusual structure type, application of "traditional" ionic models typically used to achieve rationalisation of crystal structures and why they exist does not provide useful information here. The analysis of the magnetic properties in terms of a crystal field model for the Fe^{4+} centres achieves its goal, but the statement that the crystal field splitting provides the energetic driving force for stabilisation of the crystal structure is less convincing.

Our response: We did not intend to argue that the crystal field splitting provides a stabilization of the crystal structure. Our remark in the conclusion was based on a comparison with CaFeO_3 and related perovskites, where the octahedral ligand field surrounding Fe(IV) leads to an electronic instability, which in CaFeO_3 is alleviated by charge disproportionation ($\text{Fe(IV)} \rightarrow \text{Fe(III)}, \text{Fe(V)}$). In our Ca_4FeN_4 compound no such electronic instability exists, which is probably why we do not observe charge disproportionation. We rewrote this section to make clear that we do not think that the ligand field splitting is the reason this structure is stable but is probably just responsible for stabilisation of Fe(IV) against charge disproportionation.

I suggest that the authors first examine the extensive review of nitride crystal structures published by Nate Brese and Mike O'Keeffe as "Crystal Chemistry of Inorganic Nitrides", in the

Structure and Bonding book series by Springer Berlin, 1992, pp. 307-378 (Brese N.E., O'Keeffe M. (1992) Crystal chemistry of inorganic nitrides. In: Complexes, Clusters and Crystal Chemistry. Structure and Bonding, vol 79. Springer, Berlin, Heidelberg. <https://doi.org/10.1007/BFb0036504>). This can be a bit difficult to source and one of the authors may have to purchase the chapter, but it is well worthwhile. The review examines many different nitride crystal structures critically and systematically (even highlighting previous mistakes in crystallographic structure determinations), using the principles of "bond valence" approaches to determining and studying the oxidation number (distinct from the formal OS) combined with concepts of metal-metal 2nd-nearest neighbour interactions, to give rise to the principle of "eutaxy". In addition, the authors highlight the principle of considering metal-only sublattices with inserted (much smaller) anions, to interpret many complex structure types. Reading and considering the possibilities of applying the concepts reviewed in this work are recommended here.

Our response: Thank you for bringing this review to our attention. We added the bond valence sum calculations (see next response) and we agree that the underbonded Ca₂ and Ca₃ positions might be due to metal-metal 2nd nearest neighbour interactions as also revealed in the review, as seen in the CaGaN structure. We updated the structure discussion accordingly.

Next, using the bond valence systematics first described by ID Brown and published in an extensive table by Brese and O'Keeffe (Acta Cryst B 47 (1991) 192-197) could be very helpful to provide a more detailed picture about what is happening around the different cation sites. I have done a quick calculation using the Brese/O'Keeffe parameters for Fe-N and Ca-N bonds along with the determined bond lengths, that is very revealing. The total valence achieved at the trigonal Fe sites surrounded by N1 and N3 nitrogen atoms is 4.148, confirming the ionic model assignment to a 4+ state. The Ca₁ site achieves a valency of 2.08, close to ideal. However, the Ca₂ and Ca₃ sites are seriously underbonded (1.83 and 1.63 respectively). This can not simply be ascribed to the greater "bonding power" of the Fe atoms pulling the N atoms closer to them, because the N₂ atoms are not bonded to Fe. Instead, this most likely reflects the influence of metal-metal repulsions between the large Ca atoms sharing these N₂ centres. In fact, achieving this structure under high pressure conditions might have more to do with overcoming Ca...Ca repulsions than satisfying crystal field energetic considerations. I would urge the authors to consider these possibilities that fall outside the normal, more "comfortable" ionic view of the crystal structures and how they "work". That could make an even more valuable contribution to the overall field of solid state chemistry and structure-properties design.

Our response: We performed BVS calculations with the bond valence parameters reported by Brese and O'Keeffe. We updated the structural discussion and Supporting Information accordingly. We agree that the underbonded Ca₂ and Ca₃ atoms may be due to metal-metal interactions similar to CaGaN and included this into the discussion. We have already commented above on the influence of the ligand field in the [FeN₃]⁵⁻ anions.

Finally, the authors note correctly that the main effect of pressure might not be to increase the thermodynamic activity of nitrogen to achieve the highly nitrated compound. In fact, although the melting vs thermal decomposition relations of NaN₃ are not yet known under HPHT conditions, it is established that Ca₃N₂ acts as a stable melt component in Li₃N-Ca₃N₂-P₃N₅ compositions that readily quenched to glass after heating to ~1400°C at 1-2 GPa pressures (Grande et al Nature 369 (1994) 43).

Our response: Thank you for bringing this work to our attention. We believe that the stabilizing effect of Ca₃N₂ on the Li₃N-Ca₃N₂-P₃N₅ melts originates from the electron inductive effect, that is electrons being donated by electropositive elements to stabilize the nitride state. This effect has been investigated (DOI: 10.1016/0925-8388(92)90635-M) and often cited when stabilities of nitridometallates are discussed (e.g. DOI: 10.1016/S1359-0286(96)80091-X). Nitridophosphates are

stabilized when an electropositive atom is introduced, which can be seen from ambient pressure decomposition temperatures of highly condensed compounds, e.g. P_3N_5 ca. 850°C , $\text{LiNdP}_4\text{N}_8 > 1000^\circ\text{C}$.

It is entirely possible, and even likely, that the Ca_3N_2 - NaN_3 mixture did not decompose completely, leading to a high N_2 gaseous/fluid within the high pressure reaction chamber, that provided a constant volume environment with a suitable flux for single crystal growth.

Our response: We agree that this is a possibility as we discuss in our conclusion. However, no information is available on the high-pressure high-temperature behaviour of Ca_3N_2 - NaN_3 mixtures, which is why we are currently planning in situ experiments using synchrotron radiation to study the reaction mechanism in detail. As suggested by other reviewers we updated the discussion of the role of NaN_3 in our reaction, removing ambiguous language.

In further work, the authors should examine the thermodynamic stability of their particular crystal structure and stoichiometry relative to other binary and ternary phases, because it is highly likely that what they have produced is a thermodynamically stable structure under this combination of HPHT conditions.

Our response: We agree with the reviewer that the stability of this structure is indeed an intriguing question. We are currently considering performing additional experiments and quantum chemical computations to gain more insight into the stability and electronic structure of this compound. We, however, also agree that these are beyond the scope of this present contribution.

Reviewer #2:

The present manuscript, **NCOMMS-20-32258-T**, submitted by Kloß, Attfield and co-workers, reports, as properly titled, the "*Preparation of iron(IV) nitridoferrate Ca_4FeN_4 through azidemediated oxidation under high-pressure conditions.*" The newly synthesized, solid-state material was structurally characterized by single-crystal x-ray diffraction analysis, and the compound's electronic structure was studied by neutron diffraction and SQUID magnetization as well as ^{57}Fe -Mössbauer spectroscopy. Taken all together, the study reports a ternary nitride with a tetravalent iron ion. This reviewer appreciated reading and learning about this study, and, ultimately, would recommend publication of this work in *Nature Communications*.

Our response: Thank you for evaluating our manuscript. We revised our manuscript according to your comments as follows, which improved the quality of our contribution.

However, the present version can certainly not be recommended for publication in this journal. As is, the study is clearly written for a very specific community in the field of inorganic chemistry, namely solid-state inorganic chemistry; and hence, is of very little interest to the particularly broad readership of this journal. This is a pity as the general topic of high-valent iron species is of great interest to bioinorganic and inorganic coordination chemists, spectroscopists and computational chemists. That said, a revised version must place this work in context to those neighboring fields of inorganic chemistry!

Our response: We agree that generalizing the introduction to cover neighbouring fields of inorganic chemistry may attract a broader readership. See next responses.

Once the authors bothered to study those neighboring fields, they may find out that the iron 4+ oxidation state is not an unusual iron oxidation state *per se*.

Our response: Our discussion was focussed on solid state nitrides, for which iron 4+ is very unusual as Reviewer #1 also pointed out, and this contribution is to our knowledge the first instance of such a material that has been prepared by direct combination of starting materials and structurally elucidated. We updated the introduction to make this more clear.

In molecular coordination chemistry, K. Meyer and J. Smith reported the first structurally characterized Fe(IV) nitrides in 2008 (published in *Angewandte* and *JACS*) and the same authors published an Fe(V) nitrido complex in *Science* in 2011. Fleeting intermediates of Fe(IV), Fe(V) and even Fe(VI) nitridos were spectroscopically characterized by J. Peters (*JACS* 2004) and K. Wieghardt (e.g. *JACS* 1999, *Science* 2006). Noteworthy, I. O. Fritsky published an “indefinitely stable Fe(IV) cage complexes” in *Nature Communication*, **2016**. These studies must be mentioned and cited,

Our response: Thank you for bringing this work on nitride coordination chemistry to our attention. We updated the introduction to cover this interesting field of chemistry.

and while this reviewer acknowledges the distinctiveness of the Fe(IV) center in Ca₄FeN₄, the authors ought to tone down their “unusual” and “high” oxidation state.

Our response: This has been corrected.

Further revisions should include the following:

The abstract is badly written. Why are high oxidation states difficult to stabilize due to the high thermodynamic stability and inertness of dinitrogen. I understand what the authors meant to say, but, as written, it makes no sense.

Our response: The phrasing has been altered.

Also, what is a “reactive nitrogen species.”

Our response: As the state of NaN₃ during reaction cannot easily be determined we removed this statement from the abstract and this phrasing from the conclusion as it could be misleading. We later discuss the state of NaN₃ in the conclusion section of the article (also see later responses).

In addition, the sentence is grammatically wrong, and the last sentence of the abstract reads bad, at best.

Our response: Both have been corrected.

On page 4, line 61, the authors state that sodium azide, NaN₃, is a “powerful solid-state nitriding agent.” However, the authors do not know where the nitride nitrogen atoms in their title compound stem from! This is a severe deficiency and ought to be addressed; possibly, by the use of 15-N-labeled NaN₃. In fact, how can the authors rule out that the nitrides in their [Fe(IV)N₃]₅₋ moiety do not originate from the precursor materials, Ca₃N₂ and Fe₂N ?

Our response: As shown in the Powder Diffraction section in SI, we have been able to make near phase-pure samples from HPHT reactions. The stoichiometric combination of Ca_3N_2 and Fe_2N does not have enough N to give Ca_4FeN_4 except by taking N from NaN_3 which is the only available N-source under the closed conditions of the reaction. ^{15}N labelling experiments would not be straightforward or useful, as exchange kinetics are fast under HPHT conditions so ^{15}N would rapidly be scrambled, and it would be difficult to separate different solids to see which contained ^{15}N in the unlikely event that it was segregated.

We have revised parts of the synthesis section and conclusion to discuss this matter more concisely. To investigate the reaction mechanism, we are currently planning in situ synchrotron diffraction experiments to study this problem in detail, but as this is beyond the scope of this present contribution, we maintain our working theory as given in the conclusion.

On page 4, line 66, the authors continue and state that “Sodium, which is released upon reaction, acts as a metallic flux...” This implies that NaN_3 is thermally decomposed to Na and N_3 radicals. The latter recombine and decompose to N_2 . This is literature-known and should be mentioned here.

Our response: We do not know if NaN_3 is thermally decomposed or the azide reacts directly. We rewrote this section and give a more detailed discussion in the synthesis section and the conclusion.

Whether or not the high-pressure conditions change this decomposition pathway is purely speculative.

Our response: We agree that the conclusion read too speculative and ambiguous. We rewrote this section to state our intentions more clearly. We believe that the question of the state of NaN_3 is very important as this could differentiate high-pressure synthesis in multianvil presses from medium pressure synthesis using sodium fluxes and N_2 as oxidizing agents.

Throughout the text, and the SI, when talking about Moessbauer spectroscopy, the authors must specify the isotope. Clearly, it is “ ^{57}Fe -Moessbauer” spectroscopy.

Our response: This has been updated.

Further, in Moessbauer spectroscopy, it is not the “chemical shift” (e.g. page 5, line 90) but the “isomer shift.”

Our response: Done.

Page 6, line 106/107: the “low-spin configuration” should be further specified as $S = 1$.

Our response: Done.

Page 6, line 115: the individual and average Fe–N bond distances should be mentioned/listed in the main text; e. g., page 6, line 115.

Our response: Done.

Page 8, line 143: The authors report the Weiss constant and the effective moment, μ_{eff} , to be $3.08(1) \mu_{\text{B}}$, and state that “the paramagnetic moment is close to the theoretical spin-only value of a system with $S = 1$ of $2.83 \mu_{\text{B}}$ and hence corroborate the low-spin state in the...” This reviewer would appreciate a plot of μ_{eff} vs. T , and wonders

Our response: The paramagnetic magnetic moment is independent on temperature in the temperature region (80-300K) where the Curie-Weiss law applies. We added an insert to Figure 2 to show that the inverse susceptibility plot is linear in the plotted region.

a) how the experimental error “(1)” is determined,

Our response: This is the error of the Curie-Weiss fit, not incorporating a Fe impurity.

b) and how can the authors make this statement, if the ^{57}Fe -Mössbauer spectrum reveals an obvious and significant Fe-impurity (which, very likely, will affect the effective magnetic moment).

That said, the Fe-impurity (Fig. 3) must be quantified !

Our response: We added the quantification of the Fe-impurity in the Mössbauer spectra. Owing to strong signal overlap the ratio might not be correctly determined. As the signals appear at a more positive isomer shift, it is likely a lower oxidation state of Fe thus probably increasing the observed magnetic moment, which is what we observe. Correction of the observed magnetic moment is difficult as the byproduct is unknown and hence its contribution to the magnetic moment. However, neutron powder diffraction shows a magnetic moment corresponding to a saturated moment of $2S = 2 \mu_{\text{B}}$. We added this to the discussion of the magnetic moment.

Reviewer #3:

The manuscript presents a new synthetic method for preparation of transition metal nitrides which enables formation of high oxidation states on the metal. A new compound is made in this manner and characterized using magnetic susceptibility, Mossbauer spectroscopy, and X-ray and neutron diffraction to determine its structure and the nature of the iron. This work is of interest to a broad audience as it covers a new synthesis technique as well as the properties of a new complex material. I would recommend publication if the following issues are addressed.

Our response: Thank you for evaluating our manuscript. We revised our manuscript according to your comments as follows, which improved the quality of our contribution.

1. The paper repeatedly highlights the fact that the use of the large volume press allows for formation of fairly large quantities of product. But the amounts of reactants used and product obtained are not mentioned anywhere in the manuscript (not even in supporting information). In order to get an idea of how advantageous this method is, the reader needs to know more about the scale of your reactions!

Our response: We added a statement that each experiment yields 50 mg of product, which compared to diamond anvil cell experiments is a lot and allows for the conduction of most physical properties' measurements.

2. The authors assume that the NaN₃ thermally decomposes to form sodium metal which acts as a metal flux. However, NaN₃ is not used in very large excess, and the reaction temperature is well above the boiling point of sodium at atmospheric pressure; is anything known about the state of sodium at 1200 C and 6 GPa?

Our response: We could not find any information regarding the state of sodium under these conditions. We modified the statement to indicate an assumption. In the discussion we only state that the reaction resembles a Na flux reaction as prolonged dwell times significantly facilitate larger single-crystal growth, which under high-pressure conditions usually point toward a flux/mineralizer.

No sodium metal is seen in PXRD of the product but the authors state that the EDS shows "finely dispersed" sodium in the sample. Please add more information to this section--is this sodium in the elemental form, or is it part of a compound ("Na-Fe-N byproduct")?

Our response: We have no indication for elemental sodium neither through powder diffraction nor through EDX spectroscopy. Only very little sodium is used in the reaction, ca. 5.5 at-% of the solid reaction products, and sodium is a weak X-ray scatterer compared to the main product Ca₄FeN₄. Hence, high-resolution transmission electron microscopy would probably be needed to determine if sodium is finely dispersed as a metal or is constituent of the unknown byproduct. Further follow on studies such as our planned in situ synchrotron experiments will be needed to address this point. We added a shortened version of this discussion to the corresponding section.

REVIEWER COMMENTS

Reviewer #1 (Remarks to the Author):

I believe that the authors have responded to all of the referee comments positively and that their changes have enhanced the appeal of the manuscript to a wide audience of Nature readers. I suggest that the manuscript can be published without further change.

Reviewer #2 (Remarks to the Author):

The present manuscript, NCOMMS-20-32258-A, submitted by Kloss, Attfield and co-workers, is a rebuttal of NCOMMS-20-32258-T, titled, the "Preparation of iron(IV) nitridoferrate Ca_4FeN_4 through azide-mediated oxidation under high-pressure conditions."

The authors have addressed some of this reviewer's previous concerns and suggestions but – again – have not addressed the most critical point; namely, the significant (!) Fe impurity, which impedes a proper analysis of the magnetization data, and, consequently, doubts the compound's identity and the interpretation of its electronic structure.

In the first report, this reviewer stated:

"Page 8, line 143: The authors report the Weiss constant and the effective moment, μ_{eff} , to be $3.08(1) \mu\text{B}$, and state that "the paramagnetic moment is close to the theoretical spin-only value of a system with $S = 1$ of $2.83 \mu\text{B}$ and hence corroborate the low-spin state in the..."

This reviewer would appreciate a plot of μ_{eff} vs. T , and wonders

- how the experimental error "(1)" is determined,
- and how can the authors make this statement, if the ^{57}Fe -Mössbauer spectrum reveals an obvious and significant Fe-impurity (which, very likely, will affect the effective magnetic moment).

That said, the Fe-impurity (Fig. 3) must be quantified !"

Besides the fact that the μ_{eff} vs. T plot is still not shown in this rebuttal, which is a minor criticism, the "area ratio Fe^{4+}/Fe impurity" is now given in the caption to Fig. 3; and it is stated that the "area ratios may not be highly accurate." This is not acceptable. The impurity species can be reasonably well fitted to the experimental spectrum, and while this reviewer agrees that the isomer shift and line widths is a function of temperature, and thus varies, the area ratio (the amount of impurity) cannot vary between 6 – 20% !

On this note, and with regards to the original comment above, if a sample contains up to 20% Fe impurity, which is not "a small [...] impurity" (according to p.7, line 141), it is impossible that "the paramagnetic moment is close to the theoretical spin-only value of a system with $S = 1$ of $2.83 \mu\text{B}$ and hence corroborates the low-spin state in the..." (p. 7, line 148/149).

Finally, and as mentioned above, the isomer shift is a function of the temperature (due to the temperature dependence of the Debye-Waller factor); however, the authors do not address the fact that Fe impurity and Fe target possess an opposite temperature dependence, which is unusual.

In conclusion, the magnetization and Mössbauer data leave too many open questions and do not support a central piece of information regarding the identity of the newly claimed material.

In general, if an author claims a novel "synthesis [that] opens a way to the discovery of a new class of highly oxidised nitrides", which, in fact, is not a "class of compounds" but a "single compound", the title material ought to be *pure* and *thoroughly analyzed*. This is not the case

here.

Accordingly, this reviewer cannot recommend this study for publication in Nature Communications.

Responses to comments by Reviewer #2

Reviewer #2 (Remarks to the Author):

The present manuscript, NCOMMS-20-32258-A, submitted by Kloss, Attfield and co-workers, is a rebuttal of NCOMMS-20-32258-T, titled, the “Preparation of iron(IV) nitridoferrate Ca_4FeN_4 through azide-mediated oxidation under high-pressure conditions.”

The authors have addressed some of this reviewer’s previous concerns and suggestions but – again – have not addressed the most critical point; namely, the significant (!) Fe impurity, which impedes a proper analysis of the magnetization data, and, consequently, doubts the compound’s identity and the interpretation of its electronic structure.

Our response: We address the sample impurity issue now with a specific section (7) in the Supporting Information, and other analysis throughout. It is important to note that multiple (sometimes combined) high pressure samples have been used in this study, so ‘the Mossbauer sample’ is not identical to ‘the magnetisation sample’ as far as impurities are concerned. We apologise that this was not made clear before. Furthermore, Ca_4FeN_4 is highly air and moisture sensitive (lifetime < 10 sec in air) so sample decomposition may account for the high impurity concentration seen in the Mössbauer sample in particular as it required shipping from Europe to Japan. However, the major magnetic features in the susceptibility, magnetic neutron and Mössbauer data may be assigned to Ca_4FeN_4 with high confidence, despite the presence of unidentified secondary phases, as we now make more clearly in the manuscript and SI (detailed comments below).

In the first report, this reviewer stated:

“Page 8, line 143: The authors report the Weiss constant and the effective moment, μ_{eff} , to be 3.08(1) μB , and state that “the paramagnetic moment is close to the theoretical spin-only value of a system with $S = 1$ of 2.83 μB and hence corroborate the low-spin state in the...”

This reviewer would appreciate a plot of μ_{eff} vs. T , and wonders

- how the experimental error “(1)” is determined,
- and how can the authors make this statement, if the ^{57}Fe -Mössbauer spectrum reveals an obvious and significant Fe-impurity (which, very likely, will affect the effective magnetic moment).

That said, the Fe-impurity (Fig. 3) must be quantified !”

Our response: We have now introduced a plot to show T variation of the effective moments from Curie-Weiss and Curie functions in the Supporting Information as Fig. S7. The former is constant with T over a wide T range (75 – 300 K) which would not occur if any large % of another paramagnetic phase were present (as they would have different theta values, giving a non-linear variation of inverse susceptibility). The Curie law μ_{eff} shows that the Curie tail is equivalent to an effective moment of no more than 0.3 BM, equivalent to 3% $S = \frac{1}{2}$ or less for larger spins (e.g. only 0.3% of a high spin Fe^{3+} impurity). The M-H loops show that only a trace of ferromagnetic impurity is present, equivalent to $\sim 0.01\%$ Fe metal (Fig. S6). Taken together, these results indicate that the secondary phases observed in the powder diffraction data are mainly non-magnetic, in keeping with the 4:1 Ca:Fe ratio of metals and also the presence of some Na in the bulk sample.

(a) Error ‘(1)’ is the fitting error (estimated standard deviation) from the least squares fit, as used elsewhere in the paper for crystallographic results and their errors.

(b) See SI and earlier response; the high impurity signal in Mössbauer is from a different, and probably slightly decomposed, sample to the sample used for magnetisation measurements.

Besides the fact that the μ_{eff} vs. T plot is still not shown in this rebuttal, which is a minor criticism, the “area ratio Fe^{4+}/Fe impurity” is now given in the caption to Fig. 3; and it is stated that the “area ratios may not be highly accurate.” This is not acceptable. The impurity species can be reasonably well fitted to the experimental spectrum, and while this reviewer agrees that the isomer shift and line widths is a function of temperature, and thus varies, the area ratio (the amount of impurity) cannot vary between 6 – 20% !

Our response: We agree and we have refitted the Mössbauer spectra accordingly. An area of 14.2 % for the Fe impurity is now determined from the room temperature spectrum and fixed for the low temperature spectra. This gives a more stable fit for the low temperature data, but it does not significantly change the derived Mössbauer parameters for the Ca_4FeN_4 phase which remain consistent with the other data.

On this note, and with regards to the original comment above, if a sample contains up to 20% Fe impurity, which is not “a small [...] impurity” (according to p.7, line 141), it is impossible that “the paramagnetic moment is close to the theoretical spin-only value of a system with $S = 1$ of $2.83 \mu\text{B}$ and hence corroborates the low-spin state in the...” (p. 7, line 148/149).

Our response: See above, text has been changed accordingly.

Finally, and as mentioned above, the isomer shift is a function of the temperature (due to the temperature dependence of the Debye-Waller factor); however, the authors do not address the fact that Fe impurity and Fe target possess an opposite temperature dependence, which is unusual.

Our response: Isomer shifts for the impurity doublet at low temperatures are unreliable as it is overlapped by the main signal, and we do not claim any physical significance for these numbers. The parameters for the main Ca_4FeN_4 phase and their thermal variations are physically plausible, showing that the procedure for fitting the impurity signal has not greatly affected them.

In conclusion, the magnetization and Moessbauer data leave too many open questions and do not support a central piece of information regarding the identity of the newly claimed material.

Our response: We hope that these issues are now clarified satisfactory above. We reiterate that the central magnetic features in the susceptibility, magnetic neutron and Mössbauer data may be assigned to Ca_4FeN_4 with high confidence, despite the presence of unidentified secondary phases.

In general, if an author claims a novel “synthesis [that] opens a way to the discovery of a new class of highly oxidised nitrides”, which, in fact, is not a “class of compounds” but a “single compound”, the title material ought to be *pure* and *thoroughly analyzed*. This is not the case here.

Accordingly, this reviewer cannot recommend this study for publication in Nature Communications.

Our response: Preparation of 100% pure compounds in solid state chemistry is often difficult because purification methods such as recrystallisation, washing, etc that may be used for coordination complexes, organic molecules, etc are not applicable here. It is particularly difficult to optimize synthesis in the multidimensional parameter space of high-pressure chemistry (we updated the synthesis section to give an overview of the examined parameter space) and prepare phase pure materials as each run is time-consuming and expensive, and it is usual that non-trivial new materials contain small amounts of secondary phases. This is particularly true for highly air/moisture sensitive materials like Ca_4FeN_4 . Nevertheless, we can be confident about the composition and structure of Ca_4FeN_4 (based on single crystal x-ray analysis and EDX on single-crystals), its magnetisation data (where impurity contributions are shown to be small as above), and its long range ordered magnetic structure (determined from powder neutron diffraction where the temperature and 2θ dependence of magnetic peaks enables them to be separated unambiguously from other contributions). These are the major characterisation data for the new material, which are self-consistent and also consistent with the properties expected from ligand-field theory for a trigonal-planar $[\text{FeN}_3]^{5-}$ moiety and thus constitute a proper analysis. Mössbauer parameters are consistent with these data despite the unfortunate presence of an unidentified impurity.

Although we have only prepared one material in this study, there are plenty of other (transition) metals where the highest observed oxidation state in nitrides to date is below those observed in oxides or fluorides, so there is every reason to believe that many other new high oxidation state nitrides will be accessible. There is nothing special about Fe to suggest that this chemistry would work for Fe alone. Two other expert reviewers have accepted the importance of this new method for solid state nitride synthesis and have recommended publication.

REVIEWER COMMENTS

Reviewer #2 (Remarks to the Author):

The authors addressed this reviewer's concerns in detail, solved some of the issues raised, and provided further information and explanation on the title complex's significant (14%) and unidentified impurity; mostly in the Supporting Information file. The additional note that "the Moessbauer sample" was not identical to sample used for the magnetization study was particularly helpful. In general, the data seem to be consistent now, and this reviewer recommends publication of the present study in Nature Communications. nd this study for publication in Nature Communications.